# Past and Ongoing Field-Based Studies of Myxomycetes

**DOI:** 10.3390/microorganisms11092283

**Published:** 2023-09-11

**Authors:** Steven L. Stephenson

**Affiliations:** Department of Biological Sciences, University of Arkansas, Fayetteville, AR 72701, USA; slsteph@uark.edu

**Keywords:** biodiversity, biogeography, slime molds, surveys

## Abstract

Evidence from molecular studies indicates that myxomycetes (also called myxogastrids or plasmodial slime molds) have a long evolutionary history, and the oldest known fossil is from the mid-Cretaceous. However, they were not “discovered” until 1654, when a brief description and a woodcut depicting what is almost certainly the common species *Lycogala epidendrum* was published. First thought to be fungi, myxomycetes were not universally recognized as completely distinct until well into the twentieth century. Biodiversity surveys for the group being carried out over several years are relatively recent, with what is apparently the first example being carried out in the 1930s. Beginning in the 1980s, a series of such surveys yielded large bodies of data on the occurrence and distribution of myxomycetes in terrestrial ecosystems. The most notable of these were the All Taxa Biodiversity Inventory (ATBI) project carried out in the Great Smoky Mountains National Park, the Planetary Biodiversity Inventory Project (PBI) carried out in localities throughout the world, and the Myxotropic project being carried out throughout the Neotropics. The datasets available from both past and ongoing surveys now allow global and biogeographical patterns of myxomycetes to be assessed for the first time.

## 1. Introduction

The myxomycetes (also called myxogastrids or slime molds) are a group of eumycetozoans common but often overlooked in most terrestrial ecosystems. Because many species produce fruiting bodies large enough to be observed with the naked eye, there is no doubt that they have been noticed in nature by humankind for a very long time (Figure 1). Evidence of this is found in writings from the ninth century attributed to the Chinese scholar Twang Ching-Shith, who mentioned a certain substance, *key hi* (literally “demon droppings”), that has a pale color and grows in shady damp conditions [1]. This description fits *Fuligo septica*, a common myxomycete that achieves a size that makes this species readily conspicuous. However, the first reference of a myxomycete in the published literature is probably that of the German botanist Thomas Panckow [2]. In his *Herbarium Portatile*, published in 1654, there is a brief description and a woodcut depicting what is almost certainly the common myxomycete *Lycogala epidendrum* (Figure 2).

Evidence from molecular studies indicates that myxomycetes have a long evolutionary history and probably were present on the earth several hundred million years ago. However, due to the very fragile nature of the fruiting body, fossils of myxomycetes are exceedingly rare. Domke [3] described a species of *Stemonitis,* and Dorfelt et al. [4] a species of *Arcyria* from Baltic amber dating approximately 50 million years ago, but the earliest known record of a myxomycete is a fossil of *Stemonitis* [5] from the mid-Cretaceous, about 100 million years ago (Figure 3). Interestingly, except for their age, all three fossils could easily have been assigned to still extant species.

Myxomycetes were long thought to be fungi. The fruiting bodies they produce morphologically resemble those of certain fungi (especially some small basidiomycetes and gasteromycetes), and the two groups of organisms occur in many of the same habitats. However, the fruiting bodies of fungi and myxomycetes are structurally very different, and their life cycles have little in common except that sexual reproduction involves the production of resistant spores.

## 2. Life Cycle

Myxomycetes are characterized by a relatively complicated life cycle that was not understood completely until the late 1880s [6]. In brief, the life cycle consists of two very different trophic (or feeding) stages along with a reproductive stage that bears no resemblance whatsoever to either of the trophic stages. The sequence of events in the life cycle (Figure 4) begins with a microscopic spore that formed within and then was released from the fruiting body, which represents the reproductive stage in this group of organisms. Under suitable conditions, the spore germinates to produce one to four haploid, unwalled protoplasts (or cells).

Some protoplasts become flagellated soon after being released, while others remain amoeboid. Flagellated cells are called swarm cells, while the nonflagellated cells are called myxamoebae. Myxamoebae and swarm cells are interconvertible, and the particular form in which a given cell exists apparently depends to a large extent upon the availability of free water in its immediate environment. Because swarm cells and myxamoebae are interconvertible, the general term “amoeboflagellate cell” is often used for either type of cell. In the presence of free water, the flagellated form tends to predominate, while under drier conditions, most cells exist in the nonflagellated amoeboid form [7].

Under conditions unfavorable for continued growth or metabolism, amoebo-flagellate cells undergo a reversible transformation to dormant structures called microcysts. Microcysts can remain viable for long periods of time and are probably very important in the continued survival of myxomycetes in some habitats.

The single most distinctive feature of the myxomycete life cycle is the plasmodium [8]. Indeed, myxomycetes are often referred to as plasmodial slime molds. The plasmodium (Figure 5) itself is a free-living multinucleate mass of protoplasm that is derived from the amoeboflagellate stage.

Plasmodia have no cell walls, but a thin layer of slime is usually present. The latter serves as protection against injury or desiccation. Plasmodia vary in color from hyaline to black, although white and yellow examples are common. Under adverse conditions such as drying out of the immediate environment or low temperatures, a plasmodium may convert into a hardened, resistant structure called a sclerotium. When favorable conditions return, the sclerotium is capable of reforming the plasmodium.

Eventually, if conditions remain favorable for its growth and development, a plasmodium undergoes a remarkable transformation into one or more fruiting bodies. It is still not known just what factors are involved in triggering the fruiting response, but it has been demonstrated that exhaustion of the available food supply and changes in moisture, temperature, and pH can induce fruiting in some species, Myxomycete fruiting bodies vary widely in color, shape, and size (Figure 6). A few species, such as *Fuligo septica* (Figure 1), can reach a considerable size. Even those species that produce moderately sized (±1.0 mm) fruiting bodies are capable of occurring in massive fruiting bodies that can sometimes exceed a meter or more in total extent. Fruiting bodies are ephemeral structures, and only the “toughest” examples are recognizable after a few weeks. Heavy rain tends to wash away fruiting bodies, and they are also subject to being colonized by filamentous fungi. However, if a specimen is air-dried and properly curated (i.e., placed in a suitable storage container and carefully maintained), it does not degrade like specimens of fleshy fungi) and remains suitable for study for several centuries [9,10].

## 3. The First Specimens to Be Collected

The first specimens of myxomycete for which we have records were collected from the early eighteenth to the early nineteenth centuries by Europeans whose primary focus was on other groups of organisms, particularly fungi. Since myxomycetes were considered to be fungi at the time, it is not surprising that the records of myxomycetes appeared along with fungi in species lists and publications. Specimens collected in other parts of the world by other individuals were often sent to some of these Europeans for identification. Obviously, collecting a specimen is one thing, but knowing what you have is quite a different thing.

Studies of myxomycetes beyond collecting began with Micheli (1679–1737) in Italy. Micheli was the leading mycologist of his time, and he provided descriptions of a number of taxa that can be recognized as myxomycetes. He also appears to be the first person to use a microscope to study these organisms [11]. Linnaeus (1701–1778) included seven species of myxomycetes in his *Systema Plantarum* but not in the genera used today. He made a monumental contribution to the science of taxonomy but not the study of myxomycetes. The first published work on the myxomycetes that contained a reasonable number (in this case, ca. 30) of illustrations of myxomycetes was produced in France by Bulliard (1752–1793). The illustrations were good enough to allow someone to identify at least a few of the more common species of myxomycetes. Other Europeans who included at least some myxomycetes in their publications were Schumacher (1757–1830) in Denmark, Schrader (1787–1836) in Germany, Fries (1794–1878) in Sweden, Berkeley (1803–1847) in England, de Bary (1831–1888) in Germany, and Rostafinski (1850–1928) in Poland, who published the first monograph on the group between 1874 and 1876.

It could be argued that meaningful surveys for myxomycetes would have been difficult, if not impossible, until a comprehensive illustrated monograph on the myxomycetes became readily available. This happened with the publication of George Massee’s *A Monograph of Myxogastres* in 1892, but Arthur Lister’s *A Monograph of the Mycetozoa* in 1894 [12] was much more important. The latter book was followed by two subsequent editions, published by Arthur’s daughter Gulielma, in 1911 [13] and 1925 [14]. In the United States, Thomas Macbride published his *North American Slime-Moulds* in 1899 [15], followed by a second edition in 1922 [16]. Later, Macbride collaborated with George Martin to produce *The Myxomycetes* in 1934 [17]. This was followed by another book with the same title that appeared in 1969 [18] as a result of a collaboration between Martin and Constantine Alexopoulos. This second version of *The Myxomycetes*, which contained excellent illustrations by Ruth Allen, served as the “standard” reference on the identification of myxomycetes for much of the next half century and only recently has it been supplanted, at least in part, by more recent works.

## 4. Biodiversity Surveys

For most of the time since myxomycetes were first “discovered” by humankind, records of the group almost invariably consisted of specimens that had developed under natural conditions in the field. This changed in the 1930s when Gilbert and Martin [19] first described the moist chamber technique. The technique involves placing small pieces of substrate (bark from living trees, surface litter, and other type of plant debris) in a Petri dish (or some other similar container with a lid), adding water to the dish, leaving the latter in place for a day or so, pouring off most of the water, and then leaving what is now a moist chamber culture undisturbed. Plasmodia and fruiting bodies of myxomycetes appear in the cultures, and once they are mature, the fruiting bodies can be harvested and curated in the same manner as specimens collected in the field. Interestingly, the species of myxomycetes that appear in moist chamber cultures include some species that have never been recorded in the field. Many of these are species that produce very small fruiting bodies that are virtually impossible to detect under field conditions.

Since the introduction of the moist chamber technique, it has become an important component of surveys for myxomycetes, although some examples have been limited to field-collected specimens. In contrast, there have been other surveys in which all of the biodiversity data were generated with the use of moist chamber cultures. However, in most of the more recent major survey projects described later in this paper, both field collections and specimens obtained from moist chamber cultures were considered.

Early collectors did not necessarily use any type of plan or design for the field work they carried out, and many or most of the specimens of myxomycetes they obtained were likely to have been “chance encounters” in habitats that were favorable for these organisms. This approach can yield potentially useful information relating to phenology, habitat, and substrate, but it does not contribute to expanding what is known about overall patterns of myxomycete biodiversity or allow estimates of their biodiversity at a particular locality to be determined. However, it did provide material that could be used for studies of the biology and taxonomy of myxomycetes. Moreover, many of the more common species of myxomycetes were described by these early collectors, albeit not necessarily under the taxonomic name used today.

## 5. Multi-Year Surveys

Multi-year studies of the biodiversity of myxomycetes are relatively recent, and prior to the inclusion of specimen data obtained with the use of moist chamber cultures, the very earliest studies could not be considered truly complete. Nevertheless, even these studies contributed to what we now know about biodiversity and also paved the way for the studies that followed.

Although some individuals, especially amateurs fascinated by the color and intricate nature of myxomycete fruiting bodies, have compiled lists of the species they recorded over a period of several years, most of these were never published, and the specimens were not always deposited in a recognized herbarium. One noteworthy exception was S. H. Rawson in New Zealand, who carried out intensive collecting for myxomycetes over a three-year period during the 1930s in the region around the city of Dunedin on the South Island. He recorded 24 genera and 84 species and varieties. His work greatly expanded what was known about the biodiversity of myxomycetes in New Zealand. However, it should be noted that all of his records were based on specimens collected from the field [20].

One of the first truly comprehensive efforts to document the myxomycetes of a particular locality was carried out by the author at the University of Virginia Mountain Biological Station. For five consecutive years during the period from 1982 to 1986, both specimens collected from the field and those appearing in moist chamber cultures were recorded for five study areas located along a moisture gradient. Each of the five study areas was representative of a different forest type, and numerous environmental parameters (e.g., composition of the vegetation, substrates available, moisture levels, temperate, light levels, soil pH, and nutrient levels) were measured or determined. The project yielded more than 1700 specimens of myxomycetes collected in the field and approximately 1500 specimens from moist chamber cultures. In addition to assessing patterns of abundance and biodiversity among the five study areas, the concepts of gradient analysis, niche breadth, and niche overlap were applied to myxomycetes for the first time [21,22].

In December 1998, at a meeting held in Gatlinburg, Tennessee, a proposal was made and then agreed upon to carry out what was to be called the All Taxa Biodiversity Inventory (ATBI) project in the Great Smoky Mountains National Park in the United States [23]. The goal of the project is to carry out a complete inventory of all life forms in the park, which has a total area of more than 2000 km^2^. Elevations in the Great Smoky Mountains National Park extend from 270 to 2025 m, and annual precipitation varies from about 140 cm at lower elevations to more than 220 cm at the very highest elevations. These two factors, coupled with the diverse topography of the park, produce an exceedingly wide range of vegetation types. Indeed, the Great Smoky Mountains National Park is considered the most biodiverse region of its size in temperate North America. Since the inception of the ATBI, a total of 21,600 different species have been tallied, about half of which were new to the park. More than 1000 of the latter were species new to science. Numerous individuals, including students, citizen scientists, faculty members, and foreign scientists, have participated in the ATBI project. Prior to the ATBI, 88 species were known from the park, and this has been increased to at least 230 at the time this manuscript was prepared [24,25]. Since the project is still ongoing, this number is subject to change.

Interestingly, there was an earlier effort to carry out an ATBI, to be led by Dan Janzen, in the Guanacaste region of Costa Rica. Although the project was abandoned for financial and political reasons, it set the stage for the ATBI in the Great Smoky Mountains National Park.

Because myxomycetes have no economic value to mankind, are not pathogenic, and the vast majority of people are unfamiliar with them, they have attracted very little in the way of funding, especially for field-based surveys [11]. However, this changed in 1997 when the National Science Foundation of the United States funded a three-year project entitled “Studies of Neotropical Myxomycetes”, which was based at Fairmont State College (now University) in the United States. Funding for the project was extended for an additional three years in 2001. The overall objective of the project was to carry out biodiversity surveys for myxomycetes in three localities in the Neotropics. These localities were the Luquillo Experimental Forest in Puerto Rico, the Guanacaste Conservation Area in Costa Rica, and a portion of the state of Tlaxcala in eastern Central Mexico. These three localities encompassed a wide range of Neotropical habitat types, ranging from warm deserts to high-elevation cloud forests. Later, the El Eden Ecological Reserve on the Yucatan Peninsula and the Maquipucuna Cloud Forest Reserve in Ecuador were added to the project. The surveys were carried out in the same manner as those carried out for the myxomycete component of the ATBI in the Great Smoky Mountains and the earlier project based at the Mountain Lake Biological Station. These surveys yielded the biodiversity data necessary to compare the assemblages of myxomycetes associated with temperate and Neotropical regions of the world [26,27,28,29,30,31,32,33,34]. Moreover, tropical lianas, inflorescences of large tropical herbaceous plants, and what has been referred to as “canopy soil”, were investigated as new or understudied substrates for myxomycetes [35].

In 2003, two of the most ambitious survey projects for myxomycetes ever carried out received funding. The first of these was a project entitled “Global Biodiversity of Eumycetozoans”, funded by the Planetary Biodiversity Inventories Program (PBI) of the National Science Foundation and based at the University of Arkansas. The purpose of the PBI program was to provide groups of scientists with the funding required to do the fieldwork to develop a worldwide, species-level inventory of a major group of organisms. In the first year of the program, the eumycetozoans were one of the four groups of organisms selected for funding. As the name suggests, the PBI project was directed toward all three groups of eumycetozoans—protosteloid amoebae, dictyostelids, and myxomycetes. However, the myxomycetes, as the most species-rich group, were the primary focus.

The PBI project extended over six years, ending in 2009. During the project, there were numerous expeditions to countries around the world, including Argentina, Australia, Cambodia, Cuba, Guatemala, India, Kazakhstan, Kenya, Madagascar, Miramar, New Zealand, Oman, Peru, South Africa, Thailand, and the Ukraine. The one thing all of these had in common was that they represented or contained areas where myxomycetes were an understudied group. On most of these expeditions, samples were also collected for the isolation of dictyostelids and protosteloid amoebae in the laboratory.

More than fifty individuals participated in the PBI project, ranging from students (both undergraduate and graduate) to senior faculty and parataxonomists from the country in which surveys were being carried out. Many thousands of specimens were recorded or collected, providing the basis for more than one hundred presentations, book chapters, and papers for publication [36,37,38,39,40]. Moreover, introductory workshops on myxomycetes were presented to various groups of individuals, particularly students, in a number of places worldwide.

The second major project that began in 2003 was the Myxotropic project based at the Madrid Botanical Garden, which was directed by Carlos Lado and funded by the Spanish government. The objective of the Myxotropic project is similar to that of the Global Biodiversity of Eumycetozoans project, except that the region of study was limited to the Neotropics. The fieldwork involved in the project took place in a series of seven stages in which primary emphasis was directed toward particular regions or ecosystems in the Neotropics. For example, the first phase, which took place during the first three years of the project, involved surveys of the Tehuacán-Cuicatián and Sierra Gorda Biosphere Reserves, located in Mexico. These contain the southernmost deserts in North America. Later phases centered on such regions as the Monte (Argentina) and Atacama (Chile) deserts in the Southern Hemisphere, the Patagonian steppes, the coastal deserts of Peru, and the high-elevation ecosystems of the Andean Peru. The Mycotrophic project is projected to extend through 2025. Overall, the Myxotropic project has yielded the largest assemblage of records and specimens of myxomycetes available for any single region of the world [41,42,43,44,45,46]. In addition, other products of the project have included a series of papers on the taxonomy of species of myxomycetes collected during the fieldwork, especially the special ecological group of succulenticolous (“cactus loving”) myxomycetes associated with succulent plants in xeric habitats [47,48,49].

Another project that began in 2010 and is still ongoing is the “Taxonomic and ecological diversity of the mycobiota of tropical forests in Vietnam”, based at the Komarov Botanical Institute of the Russian Academy of Sciences and funded by the Ministry of Education and Science of the Russian Federation. The actual fieldwork carried out in Vietnam is based at the joint Russian–Vietnamese Tropical Research and Technology Center in Vietnam. The overall project also encompasses lichens and fungi, but Yuri Novozhilov is the supervisor for the myxomycete component. The primary objective of the project, similar to the projects already mentioned, is to characterize the biodiversity of myxomycetes in the lowland and montane tropical forests in nature preserves and national parks throughout Vietnam. The latter included the monsoon tropical forests Dong Nai Biosphere Reserve (which included Cat Tien National Park and the Vibh Cuu Nature Reserve), Bidoup Nui Ba qbe, and Chu Yang Sin National Parks. Specimens are being collected from the field and from substrate samples used to prepare moist chamber cultures. The set of data being compiled exceeds that available for any country in the tropical regions of the world [50,51,52,53,54].

Some worthwhile surveys for myxomycetes have been carried out without the support provided by research grants. Among the best examples is the survey of the Seychelle Islands carried out by Alain Michaud from France and Tetiana Kryvomaz from the Ukraine. There is relatively little data on the assemblages of myxomycetes found on isolated oceanic islands. During the period of 2011 to 2016, these two individuals carried out extensive collecting on the Seychelle Islands. Both specimens that had fruited in the field under natural conditions, as well as samples of substrates for the preparation of moist chamber cultures, were collected. Their efforts yielded a total of 143 species and infraspecific taxa of myxomycetes, representing 6 orders, 12 families, and 29 genera. Furthermore, 16 species were found to be common, 21 were found occasionally, 37 were rare, and 46 were represented by only a single record. Elevation appeared to have some effect on the distribution of species on Mahé Island, with compositional differences noted for the assemblages recorded at lower, middle, and higher elevations. Rare species collected during their surveys included *Cribraria pachydictyon*, *Echinostelium paucifilum*, and *Physarum atroviolaceum* [55,56,57].

Other islands or island groups that have been surveyed for myxomycetes include Ascension Island [58], the Auckland Islands [59], Balearic Islands [60], Campbell Island [59], the Canary Islands [61], Christmas Island [62], Cocos Island [63], the Galápagos Islands [64], the Hawaiian Islands [65,66], La Réunion Island [67], Macquarie Island [68], Norfolk Island [69], and Stewart Island [70]. There are various other islands for which there are a few records, but there are hundreds of other islands throughout the world for which no records exist.

Island biogeography as it relates to myxomycetes is seriously understudied. Stephenson [71] compared the myxomycetes recorded from Christmas Island in the tropics with those recorded from Macquarie Island in the subantarctic. Sixty-eight species had been reported from Christmas Island [58] and twenty-six species from Macquarie Island [64]. However, the two islands shared only seven species in common, and several of these were problematic determinations. Clearly, the harsh environmental conditions that exist on Macquarie Island have a major limiting factor on myxomycetes. Interestingly, two of the myxomycetes recorded from Macquarie Island and one of those from Christmas Island were new to science.

There is a special ecological group of myxomycetes found fruiting in late spring and early summer along the edges of melting snowbanks in alpine regions of the world, such as those of the Alps in Europe and the Rocky Mountains of the western United States [72]. In 1914, Charles Meylan called them “espèces nivales”, but since then, they also have been referred to as “nivicolous” or “snowbank” myxomycetes. These myxomycetes have been the focus of a long-term survey that has extended over a longer period of time than any other survey for these organisms of which the author is aware. The survey began in 1988 through the efforts of Marianne Meyer in France. Every year since then, except for 2020, when the COVID lockdown prevented it from taking place, myxomycete enthusiasts gather at pre-selected localities, such as Saint-Paul-sur-Isère and Lanslebourg-Mont-Cenis in Savoie, Luz-Saint-Sauveur in the Pyrénées in France, Bagni di Vinadio and Sampeyre in Italy, where there is easy access to snowbank habitats. Over the years, there have been more than 400 participants from at least 26 countries. Almost all of the better-known individuals who work with myxomycetes have attended at least one year. As a result, the nivicolous myxomycetes of the Alps are exceedingly well known, and a large number of specimens have been collected. Some of these have contributed to studies that have revised some of the taxonomic concepts used for myxomycetes. Seventy-two of these nivicolous species are illustrated by Michel Poulain in the book *Les Myxomycètes* [73], including 25 species of *Lamproderma*, eight species of *Meriderma*, and four species not yet conclusively identified.

## 6. Myxomycete Biogeography

As a result of the biodiversity surveys described above, a considerable body of data on the distribution and occurrence of myxomycetes in terrestrial ecosystems now exists. To what extent have these data been used, and what else can be done? Large-scale biogeographical comparisons of the assemblages of myxomycetes in two or more widely separated regions of the world or in two different types of ecoregions are still few in number.

In what appears to be the first effort to compare the assemblages of myxomycetes on different continents, Stephenson et al. [74] used a total of 3788 records of myxomycetes collected (both in the field and from moist chamber cultures) during the period of 1954–1990 from two regions of the mid-Appalachians of eastern North America and two regions of India to analyze differences in the assemblages of species present. The two regions in eastern North America and one region in India were in the temperate zone, whereas the second region in India was in the tropics. The coefficient of community indices calculated for all pairwise combinations of the four regions indicated that the one in the tropics (southern India) had the lowest overall similarity to the others. Moreover, there were differences among the four regions in the relative proportions of the members of the traditional orders of myxomycetes. For example, members of the order Physarales made up 63% of all records, much higher than the values for the other orders. Differences among the regions also existed for the substrates (woody versus non-woody) upon which fruiting occurred.

One of the first efforts to characterize the myxomycetes of an entire ecoregion was by Stephenson et al. [75], who assembled a dataset consisting of 943 records from the field and 1043 from moist chamber cultures for the assemblage of myxomycetes associated with high-latitude (essentially beyond the northern tree line) regions of the entire Northern Hemisphere. The records were obtained during the period of 1989 to 1996 and consisted of specimens collected by the three coauthors as well as records available in the literature [76,77,78]. They indicated that the myxobiota associated with high-latitude regions is a depauperate version of that found in arctic and subarctic regions directly to the south. However, a few species elsewhere recorded as rare were found to be relatively common.

There have been two other recent and somewhat similar studies. Dagamac et al. [79] compared the datasets of myxomycetes obtained from surveys carried out in paired lowland/highland regions of the Neotropics (Ecuador and Costa Rica) and the Paleotropics (Vietnam and Thailand/Philippines). All of the surveys were carried out in areas of relatively homogenous vegetation consisting of natural or near-natural forests. The ca. 7500 specimens came from both field collections and the use of moist chamber cultures. The results of the multivariate analysis indicated that geographical separation (Neotropics versus Paleotropics) explained the observed differences in the composition of myxomycete assemblages better than habitat differences (lowland versus highland).

In the other recent study, Rojas and Stephenson [80] used data from well-studied boreal (Denali National Park in Alaska), temperate (the Great Smoky Mountains National Park in the Southern Appalachians), and tropical (the La Selva Biological Station in Costa Rica) regions to assess compositional differences in the assemblages of myxomycetes associated with each region. The entire dataset used in their analyses consisted of 3558 specimens collected during the period 1963–2013 and representing 208 species were used in their analyses. The coefficient of community indices calculated for pairwise combinations of the three datasets was lowest for the tropical–boreal pair (0.38) and highest for the temperate–boreal (0.56) dataset. Among the other attributes examined for the three datasets were such things as taxonomic diversity and numbers of unique genera and species.

## 7. Looking toward the Future

In addition to the biodiversity surveys described above, there are individuals and their graduate students at a number of universities who are moving into the next phase of such studies. At the University of Costa Rica, Carlos Rojas is engaged in ongoing ecological studies of myxomycetes in the northern Neotropics, including biodiversity surveys in the understudied countries of El Salvador, Honduras, and Columbia. Some research has been directed toward niche modeling of myxomycetes in tropical forests, the assemblages of myxomycetes associated with urban landscapes, and assessing the influence of climate on the production of fruiting bodies. The latter is a three-year project carried out in Costa Rica that is designed to obtain data on phenology and the relationship between the dynamics of a forest community and the production of fruiting bodies [81,82,83,84,85,86,87].

In China, the major center for past and ongoing studies of myxomycetes is Jilin Agricultural University in Changchun. Yu Li has mentored a large number of students who have carried out field surveys on the biodiversity of myxomycetes throughout the country. As a result of these efforts, the total number of species recorded for China now exceeds 450, which is perhaps the highest for any country in the world. Many of the students have directed their research toward taxonomy, and the specimens collected from the biodiversity surveys have yielded an appreciable number of species new to science.

The primary targets for biodiversity studies in China have gradually expanded from the temperate zone in northeastern China to the entire country. However, due to the relatively limited scope of previous studies, there are still a large number of areas (e.g., Tibet, Qinghai, Hainan, and Xinjiang) that warrant additional attention. With the expanding research on the systematics of myxomycetes and the development of high-throughput sequencing technology, the efforts to assess biodiversity are no longer limited to traditional field-based collecting and the use of moist chamber cultures [88,89,90,91,92,93,94].

Thomas Edison E. dela Cruz of the University of Santo Tomas in the Philippines has been the focal point for studies of myxomycetes in his country since 2009. When he began his work, the only recent paper dealing specifically with these organisms had been a listing of the 107 species known from the Philippines as of 1981. Since then, he and his graduate students have carried out ecological studies and biodiversity surveys at a number of localities. Their efforts have increased the number of species reported from the country to 159. The Philippines consists of more than 7500 islands of various sizes and distances from neighboring islands. These islands form an archipelago that extends between latitudes 4° and 21° N.

Habitats investigated for myxomycetes in the Philippines have included limestone forests, grasslands, agricultural plantations, mangrove forests, and the ecosystems of individual islands. For example, their studies have shown that the Philippine limestone forests possibly support a unique assemblage of myxomycetes, specifically those taxa that have calcium carbonate as a major component of the fruiting body. Comparisons of datasets obtained from islands of varying sizes and distances from each other have been used to assess the island biogeography of myxomycetes. Another topic of interest has been the impact of major disturbances in the form of typhoons on assemblages of myxomycetes [95,96,97,98,99,100,101,102].

Martin Schnittler at the University of Greifswald in Germany began his research on myxomycete biodiversity while working with the author in the context of the “Studies of Neotropical Myxomycetes” project mentioned earlier in this paper. Later, he was an active participant in both the ATBI and PBI projects based in the United States. Since arriving at the University of Greifswald in 2002, he has carried out systematic surveys (combining specimens collected in the field with those from moist chamber cultures) all over the world but with special emphasis on regions of Eurasia, including eastern Siberia, the Taimyr Peninsula, the Lower Volga River Basin, and the Caspian lowlands. More recently, he has turned his attention to applying the concepts of modern molecular biology to the study of myxomycetes, including sequencing of independently inherited marker genes. This has led to investigations into cryptic speciation in myxomycetes and the development of the most effective molecular methods for species delimitation. This research has involved collaborations with a number of other scientists, including Anna Maria Fiore-Donno (Germany), Dmitry Leontyev (Ukraine), Oleg Shchepin (Russia), and Yura Novozhilov (Russia). The collaboration with Novozhilov and his research group at the Komarov Botanical Institute of the Russian Academy of Sciences has been especially productive. Among their findings are that very few of the traditional genera recognized for myxomycetes are natural taxonomic units, and some of the more common morphospecies are actually species complexes. As will be mentioned below, these facts add another level of complexity to what constitutes “true” species biodiversity in myxomycetes [103,104,105,106,107,108,109,110].

## 8. What Comes Next?

For the surveys described in this paper, identifications of the myxomycetes collected were made using observable morphological features of the fruiting body (i.e., the morphological species concept). It has long been known that fruiting bodies, especially for some species, exhibit a considerable degree of variability and thus may not conform completely to published descriptions. Some of this variability can be attributed to environmental factors, but beginning in the early 2000s, when the techniques of modern molecular biology were first applied to myxomycetes, it has become possible to go beyond morphology. In brief, being able to obtain and analyze genetic sequences from myxomycetes (mostly from fruiting bodies) provides exciting new insights into what constitutes a “biological species” as opposed to a morphological species (or morphospecies). The system of classification traditionally used for myxomycetes is currently undergoing changes that have already resulted in a realignment of taxa, even at the level of family and order [111,112,113,114]. Some of the variable species, when subjected to study using the new data that are now available, have turned out to be complexes of several to even numerous biological species. This is the case for the *Lycogala epidendrum*, the very first species described in the literature. The morphospecies, which is one of the most familiar of all myxomycetes, now appear to consist of numerous biological species. The same situation has been demonstrated for other morphospecies [111,112,113,114,115,116].

The biodiversity surveys described in this paper characterized the assemblage of species of myxomycetes present at the time and place each survey was carried out. As such, the data acquired would theoretically provide a baseline that would allow changes in this assemblage to be monitored if the initial sampling effort could be duplicated. Kryvomaz and Stephenson [117] speculated on how global climate change might impact myxomycetes. The three most important factors that determine the distribution and occurrence of myxomycetes are moisture, temperature, and the availability of suitable substrates for their growth and development. All three of these factors would be influenced by local or regional changes in climate. Increases in temperature and decreases in precipitation would have the potential to cause compositional changes in the assemblages of myxomycetes at a given localities and, ultimately, the ecosystems in which they occur. For example, in eastern North America, mesic high-elevation forests could be replaced by more xeric mixed oak forests that now occur at lower elevations. Although both forest types share numerous species in common, there are an appreciable number of others that are largely or completely restricted to one of the two types. Whether or not myxomycetes have the potential to serve as monitors of climate change is a question that warrants being addressed. Although the vast majority of people are unaware that myxomycetes even exist, this does not mean that these fascinating organisms do not make up an ecologically significant subcomponent of terrestrial ecosystems.

## Figures and Tables

**Figure 1 microorganisms-11-02283-f001:**
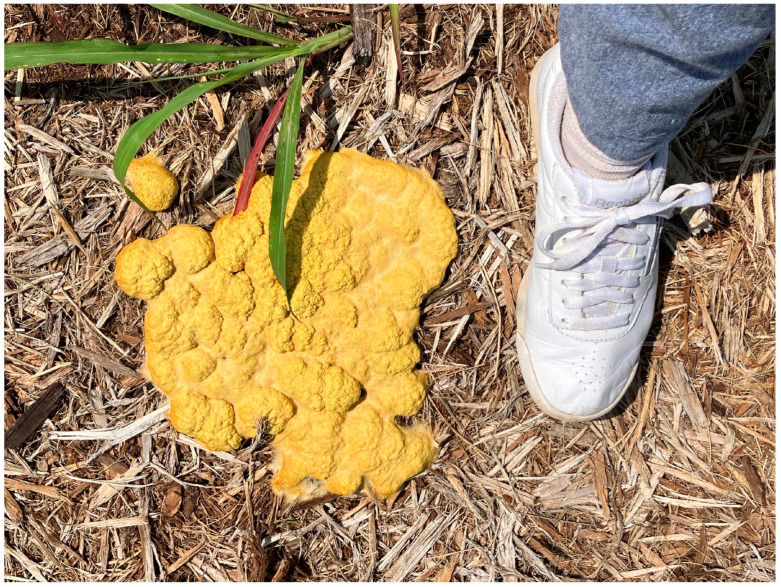
Large fruiting body (aethalium) of the myxomycete *Fuligo septica* (Barbara Stephenson).

**Figure 2 microorganisms-11-02283-f002:**
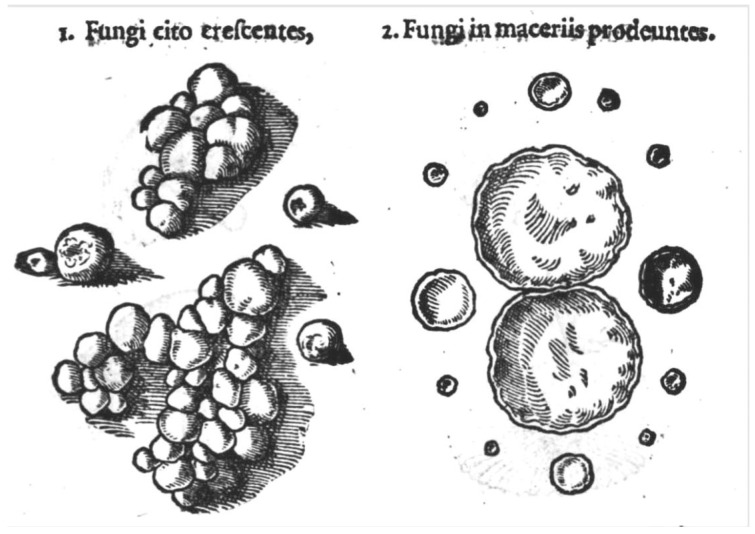
Woodcut that accompanied what appears to be the first published description of a myxomycete in Thomas Panckow’s *Herbrium Portatile* in 1654.

**Figure 3 microorganisms-11-02283-f003:**
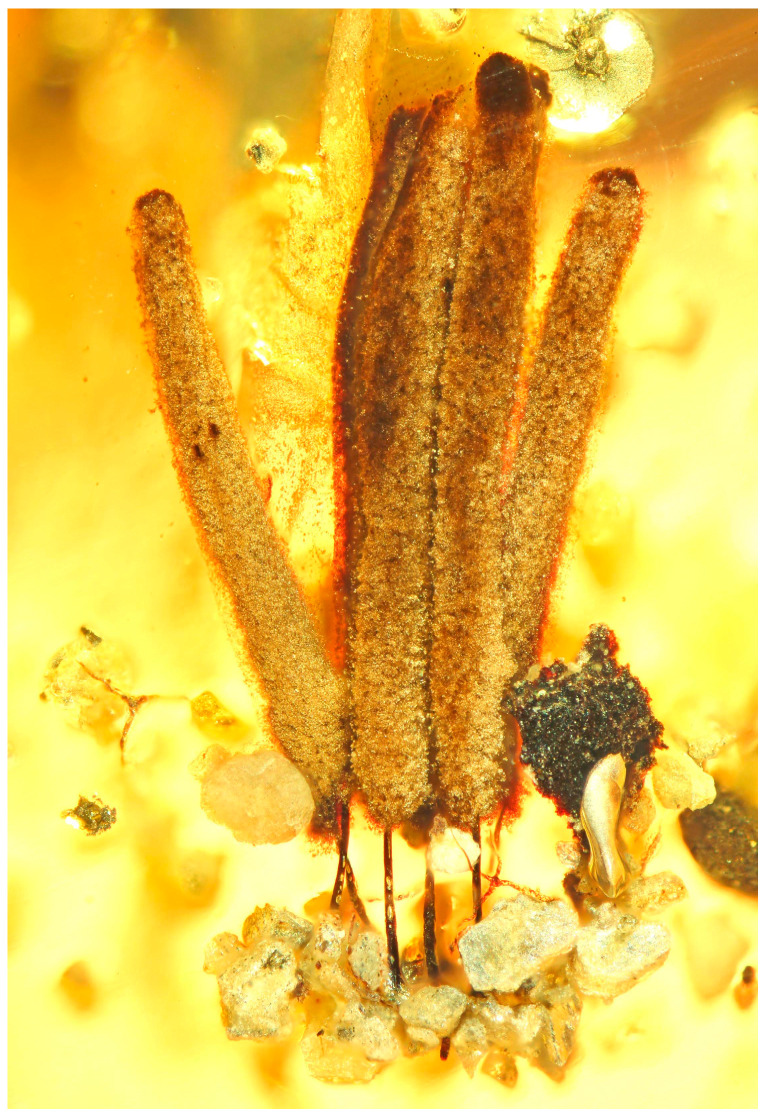
Fossil *Stemonitis* in amber preserved from the mid-Cretaceous (Alexander Schmidt).

**Figure 4 microorganisms-11-02283-f004:**
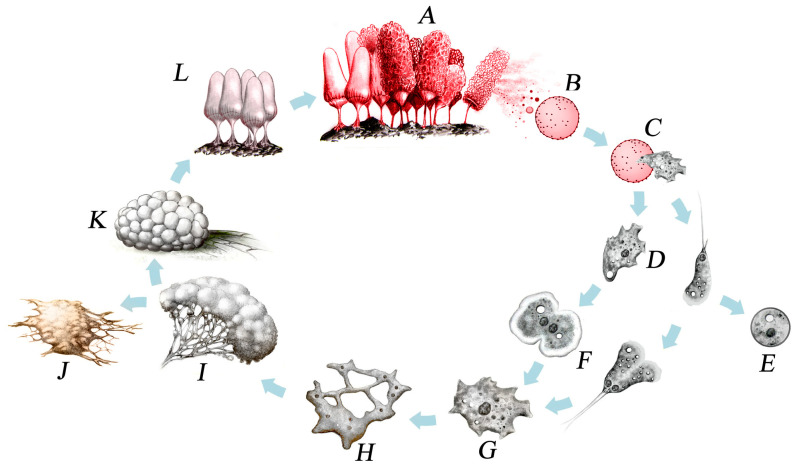
Life cycle of a typical myxomycete. A, Group of fruiting bodies. B, Spore. C, A protoplast emerges from the spore. D, The protoplast can take the form of an amoeba (left) or a flagellated cell (right) during the first trophic stage (the term “amoeboflagellate” applies to both forms). E, Under dry conditions or in the absence of food, an amoeboflagellate can form a microcyst or resting stage. F, Comparable amoeboflagellates fuse to form a zygote. G, Zygote. H, The nucleus of the zygote divides by mitosis, and each subsequent nucleus also divides without being followed by cytokinesis, thus producing a single large cell, the plasmodium. I, The plasmodium, which represents the second trophic stage in the life cycle. J, Under adverse conditions, the plasmodium can form the second type of resting stage found in myxomycetes, the sclerotium. K and L, Fruiting bodies develop from the plasmodium. During fruiting body formation, spores are produced (Dmytro Leontyev).

**Figure 5 microorganisms-11-02283-f005:**
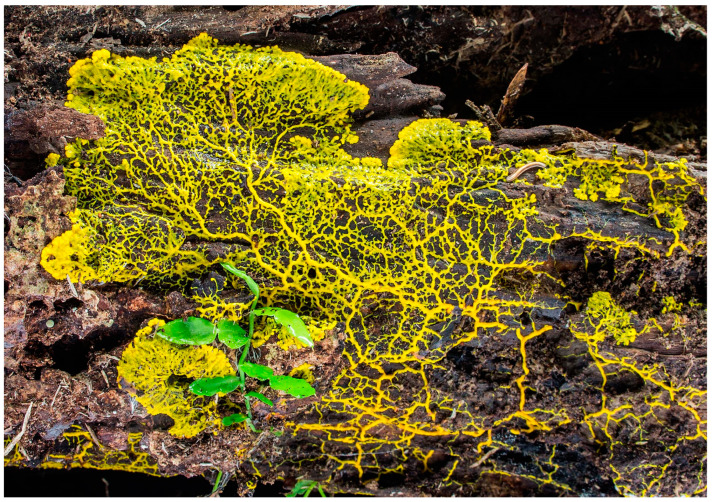
Myxomycete plasmodium (Steve Young).

**Figure 6 microorganisms-11-02283-f006:**
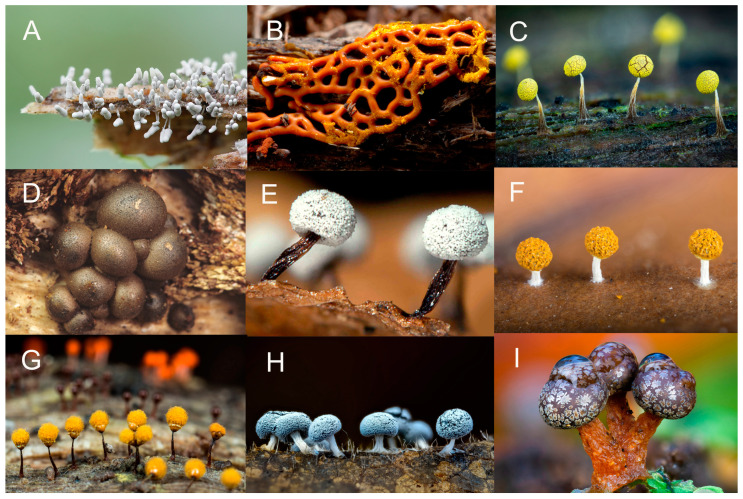
Fruiting bodies of myxomycetes (Steve Stephenson). (**A**) *Arcyria cinerea*, (**B**) *Hemitrichia serpula*, (**C**) *Physarum viride*, (**D**) *Lycogala epidendrum*, (**E**) *Didymium nigripes*, (**F**) *Physarum melleum*, (**G**) *Hemitrichia calyculata*, (**H**) *Didymium squamulosum*, (**I**) *Lepidoderma tigrinum*.

## Data Availability

Not applicable.

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
