# Peer review of "Past and Ongoing Field-Based Studies of Myxomycetes"

_microorganisms, 2023, doi:10.3390/microorganisms11092283_

Round 1

Reviewer 1 Report

The manuscript carefully reviews the state of the art on field studies in Myxomycetes, and reviews the international projects and initiatives that are being carried out with these microorganisms. The text is very well written and is understandable in all its context.

On the manuscript some minor corrections have been added (see below), and some references have been included that can help to better understand some comments. These references have been added in the corresponding paragraph but it is convenient to review their numbering so that it coincides with the references section.

It is also recommended to review and standardize the mention of journals (italics and abbreviated) in the references section.

Finally, it would be good to illustrate the diversity of shapes, colors and structures of these microorganisms, it is suggested to incorporate a plate with about 10-12 species of myxomycetes, in which the reader can appreciate the richness and plasticity of the groups or orders. Currently, only 2 species of the more than 1000 recognized are included and they do not seem enough.

Suggested changes (Green):

Line 57. (especially some small basidiomycetes and gasteromycetes)

Line 161. and Rostafinski (1850-1928) in Poland, who published the first monograph of myxomycetes between 1874-1876.

Line 164. This happened with the publication of of George Massee, A monograph of Myxogastres in 1892, and especially of Arthur Lister's...

Line 316. xeric habitats [47-49]. (see references) 

Line 348. Auckland Islands [57], Balearic Islands [58], Campbell Island [57], Canary Islands [59],

Line 512. the level of family and order [108, 111]. 

Lines 573-575. change "The MacMillian" by "The MacMillan"

Line 591. change "28" by 26

Line 610. Change "36" by 35

Line 627. Change Garcia-Carvjl by García-Carvajal

Line 633. Change Peru Part 11 by Peru Part II

Line 637. add these references and correct the numbering, also in the text.

47. Mosquera, J.: Lado, C.; Estrada-Torres, A.;  Beltrán-Tejera, E. Trichia perichaenoides, a new myxomycete associated with decaying succulent plants. Mycotaxon 2000, 75, 319-328.

48.  Mosquera, J.; Lado, C; Estrada-Torres, A.; Beltrán-Tejera, E.;  Wrigley de Basanta, D. Description and culture of a new myxomycete, Licea succulenticola. Anales Jard. Bot. Madrid 2003, 60(1), 3-10. 

49.     

Line 663. add these references and correct the numbering, also in the text

58. Lado, C; Siquier, J.L. Myxomycetes de las Islas Baleares. Catálogo de especies. 2014, Diseño e impresión Amadipesment. Palma de Mallorca, España, 2014.

59. Beltrán-Tejera, E.; Mosquera, J.; Lado, C. Myxomycete diversity from arid and semiarid zones of the Canary Islands (Spain). Mycotaxon 2010, 113, 439-442. 

Line 772. add these references and correct the numbering, also in the text

110.  García-Cunchillos, I.; Estébanez, B.; Lado, C. New approach to the ultrastructure of the capillitium in the order Trichiales (Myxomycetes) and its phylogenetic implications. Protist 172, 125805, 2021. https://doi.org/10.1016/j.protis.2021.125805.

111. García-Martín, J.M.; Zamora, J.C.; Lado, C. Multigene phylogeny of the order Physarales (Myxomycetes, Amoebozoa): shedding light on the dark-spored clade. Persoonia,  51, 89-124, 2023. DOI: https://doi.org/10.3767/persoonia.2023.51.02.

Author Response

I have added a plate of images of various myxomycetes, added the additional references, and have made the various edits suggested by the reviewer. In addition, I have checked the references section for consistency.

Reviewer 2 Report

The written review is interesting and valuable for myxomycete researchers and microbiologists. It does summarize, in one single source, quite a lot of information. The author is very well known for his work on the focal group of microorganisms and the review will likely attract attention.

For the title given and the potential expectations of readers, I think the author focuses too much on the life cycle. Perhaps if that section is shortened a little (20%?), the focal point of the paper (the studies) can be easier to keep in mind.

Right after the life cycle explanation but still within the same section, the author starts mentioning the first collections from Europeans. It is hard to follow the line of thought. Can that part begin the following section?

When talking about SH Rawson, who collected in New Zealand over the course of three years, I think it would be worth adding that his work took place in the 1930s. In the current text, there is no temporal reference for the work.

I think it would be worthwhile to focus a little more on the island surveys. In general, island biogeography is still a topic of interest for a number of scientists, and they can find this part interesting and valuable.

In section 5, the author stated that “Large-scale biogeographical comparisons of the assemblages of myxomycetes in two or more widely separated regions of the world or in two different types of ecoregions are still few in number. This is particularly interesting. Can the author provide a couple of reasons for this?

A few corrections

L7 – “… the oldest known fossil is known from…” The word known is repeated in the same sentence.

L 38 – “described” is misspelled

L157 – There is a period that should not be there. It interrupts the sentence.

L192 – “described later in this chapter”. It looks like a copy/paste from another text. Unless this paper is published as a chapter, the author should change the phrase.

L317 – there is a “i” missing in the word “in”.

Author Response

I have reduced the information provided on the life cycle, added a numbered section (as a break) after the life cycle, added the dates for Rawson, made the minor edits suggested by the reviewer, and added information to the portion of the text dealing with island biogeography.  I have no explanation as to why large-scale biogeographical comparisons of myxomycetes are limited except for the fact that few people who study myxomycetes have worked in two widely separated regions.